# GenCorres: Consistent Shape Matching via Coupled Implicit-Explicit Shape Generative Models

**Haitao Yang**[†], **Xiangru Huang**[‡], **Bo Sun**[†], **Chandrajit Bajaj** [†], **Qixing Huang**[†]
[†]The University of Texas at Austin        [‡]MIT CSAIL

## Abstract

This paper introduces GenCorres, a novel unsupervised joint shape matching (JSM) approach. Our key idea is to learn a mesh generator to fit an unorganized deformable shape collection while constraining deformations between adjacent synthetic shapes to preserve geometric structures such as local rigidity and local conformality. GenCorres presents three appealing advantages over existing JSM techniques. First, GenCorres performs JSM among a synthetic shape collection whose size is much bigger than the input shapes and fully leverages the data-driven power of JSM. Second, GenCorres unifies consistent shape matching and pairwise matching (i.e., by enforcing deformation priors between adjacent synthetic shapes). Third, the generator provides a concise encoding of consistent shape correspondences. However, learning a mesh generator from an unorganized shape collection is challenging, requiring a good initialization. GenCorres addresses this issue by learning an implicit generator from the input shapes, which provides intermediate shapes between two arbitrary shapes. We introduce a novel approach for computing correspondences between adjacent implicit surfaces, which we use to regularize the implicit generator. Synthetic shapes of the implicit generator then guide initial fittings (i.e., via template-based deformation) for learning the mesh generator. Experimental results show that GenCorres considerably outperforms state-of-the-art JSM techniques. The synthetic shapes of GenCorres also achieve salient performance gains against state-of-the-art deformable shape generators.

## 1 Introduction

Shape matching is a long-standing problem with rich applications in texture transfer (Schreiner et al., 2004), compatible remeshing (Kraevoy & Sheffer, 2004), shape morphing (Eisenberger et al., 2021), deformation transfer (Sumner & Popović, 2004), to name just a few. It also provides foundations for analyzing and processing shape collections (Kim et al., 2012; Huang et al., 2014; 2019a). As the sizes and variations of geometric shape collections continue to grow, there are fundamental challenges in formulating and solving the shape matching problems. Pairwise approaches work for similar shape pairs and become less effective on less similar shapes. The real difficulties lie in developing suitable matching potentials (that factor out usually unknown inter-shape variations) and non-convexity in the induced non-convex optimization problems.

In contrast to pairwise matching, joint shape matching (JSM) simultaneously optimizes consistent correspondences among a shape collection (Nguyen et al., 2011; Huang et al., 2012; Kim et al., 2012; Huang & Guibas, 2013; Wang & Singer, 2013; Huang et al., 2014; 2019b; 2020b;a). These techniques bypass the difficulty of matching two different shapes through paths of similar shape pairs. Despite significant advances on this topic, existing approaches present three challenges. The first is to obtain a sufficiently large dataset so that each shape has neighboring shapes where shape matching succeeds. The second is that pairwise inputs are usually detached from joint matching. Third, encoding consistent dense correspondences is costly for large shape collections.

This paper presents *GenCorres* for solving the JSM problem. GenCorres takes motivations from recent advances in neural shape generators. Given a collection of shapes with no inter-shape correspondences, GenCorres seeks to learn a mesh generator to fit the input shapes while constraining deformations between adjacent synthetic shapes to preserve geometric structures such as local rigidity and conformality (See Figure 1). Interestingly, this simple framework addresses all the challenges of JSM. GenCorres performs JSM among synthetic shapes, whose size is much larger than the number of input shapes. Second, shape matching is done among neighboring shapes through the local

rigidity and local conformality potentials, bypassing the difficulty of crafting a non-linear objective function between less similar shapes. In addition, GenCorres unifies pairwise matching (i.e., through deformation priors between adjacent shapes) and consistent matching (i.e., through the generator). Furthermore, the mesh generator provides an efficient encoding of shape correspondences.

However, learning the mesh generator directly from the input shapes is challenging as it requires good initializations. Moreover, optimization procedures, e.g., that minimize the earth-mover distances between synthetic shapes and training shapes (Fan et al., 2017; Achlioptas et al., 2018), can easily get trapped into local minimums. Gen-Corres addresses this issue by learning an implicit shape generator from the input shapes. The formulation builds on a novel approach for establishing dense correspondences between adjacent implicit surfaces defined by the shape generator. GenCorres enforces these correspondences to preserve local rigidity and conformality between pairs of adjacent shapes and satisfy the cycle consistency constraint among adjacent shape triplets. These constraints are modeled as regularization terms for learning the implicit shape generator. GenCorres then converts the learned implicit generator into an explicit mesh generator. The implicit generator offers initial consistent correspondences by guiding template-based registration.

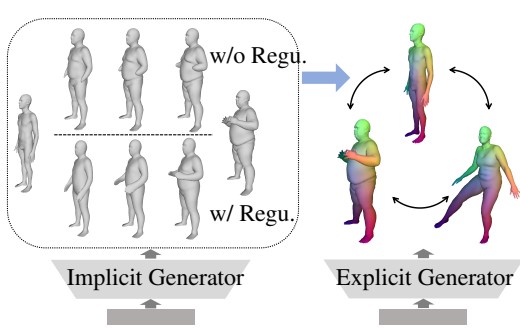

Figure 1: GenCorres performs consistent shape matching by learning a coupled implicit and mesh (explicit) generator to fit a shape collection without pre-defined correspondences. (Left) Interpolation between a pair of shape in the shape space. Constraining deformations between adjacent synthetic shapes with the regularization loss improves the shape space. (Right) The mesh generator provides consistent correspondences between pairs of shapes.

We have evaluated GenCorres on various deformable shape collections, including humans and animals. Experimental results show that GenCorres outperforms state-of-the-art JSM approaches and implicit and point cloud shape generators, making GenCorres a universal framework for computing JSM and learning deformable implicit shape generators. An ablation study justifies the importance of different components of GenCorres.

## 2 RELATED WORK

We discuss relevant work under five topics, which are described below.

**Pairwise shape matching.** Pairwise shape matching has been studied extensively in the literature (Sahillioglu, 2020; van Kaick et al., 2010; Kim et al., 2011; Ovsjanikov et al., 2012; Aigerman et al., 2014; Maron et al., 2016; Melzi et al., 2019; Bednarik et al., 2021). A recent line of papers establishes a learning framework under the functional map representation (Litany et al., 2017; Halimi et al., 2019; Donati et al., 2020; Sharma & Ovsjanikov, 2020; Cao et al., 2023). However, existing techniques still do not work well for less similar shape pairs, where it is challenging to learn suitable matching objective functions.

NeuroMorph (Eisenberger et al., 2021) combines a correspondence module with a shape interpolation module. The network is trained in an unsupervised manner. Several other methods (Eisenberger et al., 2019; Eisenberger & Cremers, 2020) also optimize interpolation paths to establish correspondences. While GenCorres is relevant to these approaches, GenCorres is a data-driven approach that uses an implicit generator with learned representations from all input shapes to drive pairwise matching.

**Joint shape matching.** The underlying principle of joint shape matching (JSM) techniques (Nguyen et al., 2011; Huang et al., 2012; Kim et al., 2012; Huang & Guibas, 2013; Wang & Singer, 2013; Huang et al., 2014; Chen et al., 2014; Huang et al., 2019b; 2020b;a) is cycle-consistency. State-of-the-art JSM techniques use the equivalence between the cycle-consistency constraint and the data matrix's low-rank property, which encodes pairwise maps in blocks (c.f. (Huang & Guibas, 2013)). This leads to constrained low-rank matrix recovery approaches (Huang & Guibas, 2013; Wang & Singer, 2013; Chen et al., 2014; Huang et al., 2017; Bajaj et al., 2018; Huang et al., 2019b), which possess strong theoretical guarantees.

GenCorres advances JSM in multiple ways. First, JSM's performance improves when the input collection size increases, as each shape can have more similar shapes for pairwise shape matching (Nguyen et al., 2011; Huang et al., 2012; Kim et al., 2012; Huang & Guibas, 2013). The advantage

of GenCorres is that it utilizes a large collection of synthetic shapes and fully leverages the data-driven behavior of JSM. Second, in prior methods, joint matching and pairwise matching are typically decoupled. CZO (Huang et al., 2020a) is an exception, yet it still requires good initializations, e.g., (Kim et al., 2011). In contrast, GenCorres unifies pairwise matching and joint matching under a simple formulation. Cycle consistency is automatically enforced through the generator. Moreover, JSM performs pairwise matching among neighboring shapes through simple geometric regularizations. Finally, JSM still requires storing consistent matches across the input shape collection (Huang et al., 2012; Kim et al., 2012; Huang et al., 2014). GenCorres addresses this issue using a shape generator to compress consistent correspondences effectively.

**Generative model based correspondences.** Generative models under explicit representations provide inter-shape correspondences, making them appealing for practical applications. However, existing methods are sub-optimal for high-fidelity correspondence computation. Most mesh-based generators (Tan et al., 2018; Verma et al., 2018; Litany et al., 2018; Tretschk et al., 2020; Rakotosaona & Ovsjanikov, 2020; Muralikrishnan et al., 2022) require consistent dense correspondences as input. In contrast to mesh-based generators, point-based generators (Fan et al., 2017; Achlioptas et al., 2018; Yang et al., 2018; Li et al., 2019a;b) do not require inter-shape correspondences. The downside is that a point cloud is permutation-invariant. Therefore, the point indices in a point cloud do not always reflect meaningful correspondences. GenCorres addresses these limitations by performing shape matching under implicit representations using shape-preserving potentials.

3D-CODED (Groueix et al., 2018) adopts an auto-encoder to deform a template shape for shape matching. The training combines the Chamfer distance for shape alignment and regularizations on Laplacian operators and edge lengths. As the regularizations of 3D-CODED are designed for isometric deformations, and the Chamfer distance drives training, it mainly applies to minor inter-shape variations. In contrast, GenCorres applies to shape collections under large deformations.

**Matching under implicit surfaces.** A fundamental problem for neural implicit shape representation is defining inter-shape correspondences. The technical challenge is that there is only one constraint along the normal direction at each surface point, c.f., (Stam & Schmidt, 2011). GenCorres solves a constrained optimization problem to obtain inter-shape correspondences. A relevant formulation has been studied in (Tao et al., 2016). Under the implicit representation, a popular way to regularize local rigidity is the Killing vector field approach (Ben-Chen et al., 2010; Solomon et al., 2011; Tao et al., 2016; Slavcheva et al., 2017), which is correspondence-free. In contrast, the correspondences computed by GenCorres allow us to introduce the cycle-consistency regularization.

**Neural implicit representations.** Neural implicit representations have received significant interest on modeling 3D shapes, including man-made objects (Park et al., 2019; Mescheder et al., 2019; Chen & Zhang, 2019; Deng et al., 2021) and deformable objects (Saito et al., 2019; 2020; Alldieck et al., 2021; Peng et al., 2021). Unlike developing novel implicit network architectures, GenCorres focuses on regularization losses that enforce geometric priors for deformable objects.

Developing regularization losses for training implicit neural networks has also been studied recently (Gropp et al., 2020; Atzmon et al., 2021). GenCorres is most relevant to (Atzmon et al., 2021), which uses an as-killing-as-possible regularization loss to preserve global rigidity. In contrast, GenCorres focuses on maintaining local rigidity and conformality.

GenCorres is also relevant to ARAPReg (Huang et al., 2021). However, defining a suitable loss under the implicit representation has to address the fundamental challenge of determining inter-shape correspondences. GenCorres also enforces the cycle-consistency constraint among induced shape correspondences to enhance the implicit generator.

## 3 PROBLEM STATEMENT AND APPROACH OVERVIEW

**Problem statement.** The input to GenCorres is a shape collection $\mathcal{S} = \{S_1, \cdots, S_n\} \subset \overline{\mathcal{S}}$, where $\overline{\mathcal{S}}$ is the underlying shape space. Each shape $S_i$ can be a raw mesh or a raw point cloud.

GenCorres seeks to learn a mesh generator $\boldsymbol{m}^\theta : \mathcal{Z} \to \overline{\mathcal{S}}$, where $\theta$ are the network parameters, $\mathcal{Z} := \mathbb{R}^d$ is the latent space. Our goal is to align each input shape $S_i$ with the corresponding synthetic shape $\boldsymbol{m}^\theta(\boldsymbol{z}_i)$ where $\boldsymbol{z}_i \in \mathbb{R}^d$ is the latent code of $S_i$. The mesh generator then provides consistent inter-shape correspondences.

**Approach overview.** As illustrated in Fig. 2, GenCorres proceeds in three stages. The first two stages provide initializations for the third stage, which learns the mesh generator. Specifically, the first stage adopts variational auto-encoder (VAE) to learn an implicit generator $f^\phi : \mathbb{R}^3 \times \mathcal{Z} \to \mathbb{R}$ and an

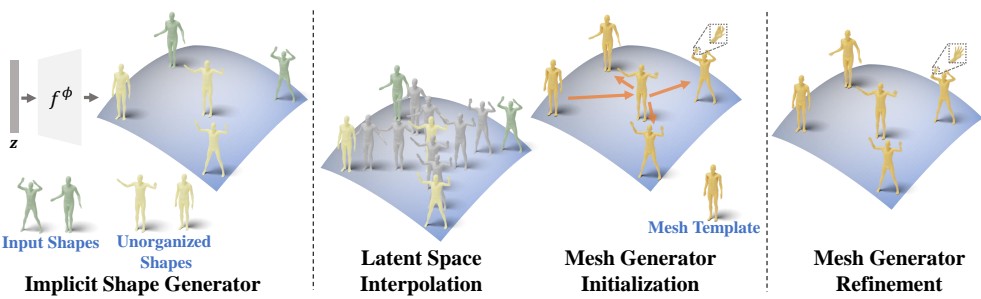

Figure 2: GenCorres has three stages. The first stage learns an implicit shape generator to fit the input shapes. The training loss regularizes the induced correspondences between adjacent implicit shapes of the generator. The second stage uses the implicit generator to initialize a mesh generator through latent space interpolation and template matching. The third stage then refines the mesh generator with ACAP energy.

encoder $h^\psi$ from the input shapes:

$$\min_{\phi,\psi} l_{\text{VAE}}\big(f^\phi, h^\psi\big) + \lambda_{\text{geo}} r_{\text{geo}}(f^\phi) + \lambda_{\text{cyc}} r_{\text{cyc}}(f^\phi) \tag{1}$$

where $\phi$ and $\psi$ are the network parameters, $l_{\text{VAE}}$ is a VAE loss on the training shapes, $\lambda_{\text{geo}}$ and $\lambda_{\text{cyc}}$ are the weights of the regularization terms. $r_{\text{geo}}(f^\phi)$ and $r_{\text{cyc}}(f^\phi)$, which are key contributions of this paper, build on induced correspondences between adjacent implicit shapes defined by $f^\phi$. Specifically, $r_{\text{geo}}(f^\phi)$ enforces that the induced correspondences preserve local geometric structures. $r_{\text{cyc}}(f^\phi)$ enforces that the induced correspondences are cycle-consistent. In other words, $r_{\text{geo}}(f^\phi)$ and $r_{\text{cyc}}(f^\phi)$ perform pairwise matching and consistent matching, respectively. The second stage of GenCorres fits a template mesh to all input shapes along paths of interpolated shapes provided by the implicit shape generator. The resulting correspondences are used to learn an initial mesh generator $\boldsymbol{m}^\theta$. The third stage of GenCorres refines the mesh generator by solving another optimization problem:

$$\min_\theta d_{\exp}\big(\boldsymbol{m}^\theta, \overline{\mathcal{S}}\big) + \lambda_{\text{d}} r_{\text{d}}(\boldsymbol{m}^\theta) \tag{2}$$

where $d_{\exp}\big(\boldsymbol{m}^\theta, \overline{\mathcal{S}}\big)$ aligns the explicit generator with the input shape collection; $r_{\text{d}}(\boldsymbol{m}^\theta)$ enforces as-conformal-as-possible deformation prior among adjacent shapes; $\lambda_{\text{d}}$ is the weight of $r_{\text{d}}(\boldsymbol{m}^\theta)$.

# 4 STAGE I: IMPLICIT SHAPE GENERATOR

This section introduces how to learn the implicit shape generator $f^\phi$. We begin with a novel approach for computing dense correspondences between adjacent implicit surfaces in Section 4.1. Based on the induced correspondences, we introduce two regularization terms $r_{\text{geo}}(f^\phi)$ and $r_{\text{cyc}}(f^\phi)$ in Section 4.2 and Section 4.3, respectively. Finally, Section 4.4 elaborates on the implementation details.

## 4.1 INDUCED SHAPE CORRESPONDENCES

Our goal is to compute the dense correspondences between the implicit surface $f^\phi(\boldsymbol{x}, \boldsymbol{z}) = 0$ and an adjacent implicit surface $f^\phi(\boldsymbol{x}, \boldsymbol{z} + \epsilon\boldsymbol{v}) = 0$, where $\boldsymbol{x} \in \mathbb{R}^3$, $\boldsymbol{v} \in \mathbb{R}^d$ is a direction in the unit ball $\mathcal{B}^d$ and $\epsilon$ is an infinitesimal value. The computation is nontrivial because of the difficulties in representing correspondences for the implicit surfaces. To this end, we first discretize $f^\phi(\boldsymbol{x}, \boldsymbol{z}) = 0$ using a mesh with $n$ vertices $\boldsymbol{g}^\phi(\boldsymbol{z}) \in \mathbb{R}^{3n}$, e.g., via Marching cube (Lorensen & Cline, 1987). We then formulate the corresponding vertices of $\boldsymbol{g}^\phi(\boldsymbol{z})$ on $f^\phi(\boldsymbol{x}, \boldsymbol{z} + \epsilon\boldsymbol{v}) = 0$ as $\boldsymbol{g}^\phi(\boldsymbol{z} + \epsilon\boldsymbol{v}) := \boldsymbol{g}^\phi(\boldsymbol{z}) + \boldsymbol{d}^{\boldsymbol{v}}(\boldsymbol{z}) \in \mathbb{R}^{3n}$. With this formulation, computing correspondences between two implicit surfaces is reduced to the computation of $\boldsymbol{d}^{\boldsymbol{v}}(\boldsymbol{z})$. As discussed in (Stam & Schmidt, 2011; Tao et al., 2016), for each vertex $\boldsymbol{g}_i^\phi(\boldsymbol{z}) \in \mathbb{R}^3$, the implicit representation offers one constraint on its corresponding $\boldsymbol{d}_i^{\boldsymbol{v}}(\boldsymbol{z}) \in \mathbb{R}^3$ along the normal direction:

$$\frac{\partial f^\phi}{\partial \boldsymbol{x}}(\boldsymbol{g}_i^\phi(\boldsymbol{z}), \boldsymbol{z})^T \boldsymbol{d}_i^{\boldsymbol{v}}(\boldsymbol{z}) + \epsilon \frac{\partial f^\phi}{\partial \boldsymbol{z}}(\boldsymbol{g}_i^\phi(\boldsymbol{z}), \boldsymbol{z})^T \boldsymbol{v} = 0. \tag{3}$$

To introduce extra constraints on $\boldsymbol{d}^{\boldsymbol{v}}(\boldsymbol{z})$, we enforce that the displacements of the 1-ring patch at each vertex $\boldsymbol{g}_i^\phi(\boldsymbol{z})$ are as-rigid-as possible (ARAP) (Alexa et al., 2000; Huang et al., 2009; 2021)

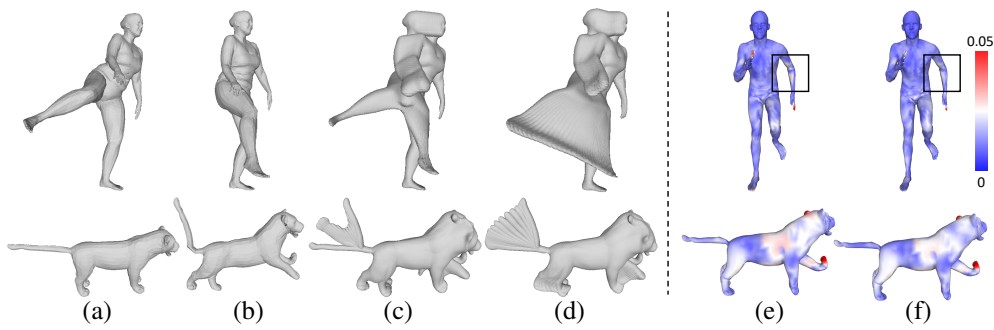

Figure 3: (Left) Effects of the geometric deformation regularization loss $r_{\text{geo}}(f^\phi)$. We compute 30 interpolated shapes between a source shape (a) and a target shape (b) via linear interpolation between their latent codes. All the interpolated shapes are visualized in the same coordinate system. (c) Interpolation results without $r_{\text{geo}}(f^\phi)$. (d) With $r_{\text{geo}}(f^\phi)$. (Right) Effects of the cycle-consistency regularization loss $r_{\text{cyc}}(f^\phi)$. We color-code errors of propagated correspondences through a path of intermediate shapes between each source-target shape pair. The error is visualized on the target mesh. (e) Without $r_{\text{cyc}}(f^\phi)$. (f) With $r_{\text{cyc}}(f^\phi)$.

and as-conformal-as possible (ACAP) (Yoshiyasu et al., 2014). In the infinitesimal regime, we can approximate the latent rotation at $g_i^\phi(z)$ as $I_3 + c_i \times$. This leads to an ARAP potential on $d^v(z)$ as

$$\sum_{i=1}^{n} \min_{c_i} \sum_{j \in \mathcal{N}_i} \left\| c_i \times \left( g_i^\phi(z) - g_j^\phi(z) \right) - \left( d_i^v(z) - d_j^v(z) \right) \right\|^2 = d^v(z)^T \overline{L}^{\text{arap}}(g^\phi(z)) d^v(z) \quad (4)$$

where the expression of $\overline{L}^{\text{arap}}(g^\phi(z))$ is in the supp. material.

Similarly, we can parameterize the latent similarity transformation at $g_i^\phi(z)$ as $(1 + s_i)I_3 + c_i \times$ and define the ACAP potential as

$$\sum_{i=1}^{n} \min_{s_i, c_i} \sum_{j \in \mathcal{N}_i} \left\| (s_i I_3 + c_i \times)\left( g_i^\phi(z) - g_j^\phi(z) \right) - \left( d_i^v(z) - d_j^v(z) \right) \right\|^2 = d^v(z)^T \overline{L}^{\text{acap}}(g^\phi(z)) d^v(z)$$

$$(5)$$

where the expression of $\overline{L}^{\text{acap}}(g^\phi(z))$ is in the supp. material.

Denote $\overline{L}^\phi(z) = \alpha \overline{L}^{\text{arap}}(g^\phi(z)) + \overline{L}^{\text{acap}}(g^\phi(z))$ where $\alpha$ is a tradeoff parameter ($\alpha = 10$ in our experiments). We compute $d^v(z)$ via linearly constrained quadratic programming:

$$d^v(z) := \lim_{\mu \to 0} \operatorname*{argmin}_d \ d^T \overline{L}^\phi(z) d + \mu \|d\|^2 \quad s.t. \quad C^\phi(z) d = -\epsilon F^\phi(z) v \quad (6)$$

where $C^\phi(z) d = -\epsilon F^\phi(z) v$ is the matrix representation of (3), $C^\phi(z) \in \mathbb{R}^{n \times 3n}$ is a block diagonal sparse matrix, $F^\phi(z) \in \mathbb{R}^{n \times d}$, $\mu$ is used to avoid degenerate cases, e.g., a rotating sphere. The expressions of $C^\phi(z)$ and $F^\phi(z)$ are in the supp. material. It is easy to check that

$$d^v(z) = -\epsilon G^\phi(z) v, \qquad G^\phi(z) := \overline{L}^\phi(z)^+ C^\phi(z)^T \left( C^\phi(z) \overline{L}^\phi(z)^+ C^\phi(z)^T \right)^+ F^\phi(z) \quad (7)$$

where $A^+$ denotes the Moore–Penrose inverse of $A$.

## 4.2 Geometric Deformation Regularization Loss

We proceed to introduce the first regularization loss $r_{\text{geo}}(f^\phi)$, which penalizes local rigidity and local conformality distortions of the induced correspondences from $f^\phi(x, z) = 0$ to $f^\phi(x, z + \epsilon v) = 0$:

$$r_{\text{geo}}(z, v) := d^v(z)^T \overline{L}^\phi(z) d^v(z) = \epsilon^2 v^T E^\phi(z) v \quad (8)$$

$$E^\phi(z) := F^\phi(z)^T \left( C^\phi(z) \overline{L}^\phi(z)^+ C^\phi(z)^T \right)^+ F^\phi(z)$$

Integrating $v$ over the unit ball $\mathcal{B}^d$ in $\mathbb{R}^d$ (Huang et al., 2021) and omitting the constant $\epsilon^2$, we define

$$r_{\text{geo}}(f^\phi) = \mathbb{E}_{z \sim \mathcal{N}_d} \int_{v \in \mathcal{B}_d} v^T E^\phi(z) v \, dv = \mathbb{E}_{z \sim \mathcal{N}_d} \frac{\text{Vol}(\mathcal{B}_d)}{d} \text{Tr}(E^\phi(z)) \quad (9)$$

Figure 3 (Left) shows that $r_{\text{geo}}(f^\phi)$ can improve the quality of the implicit shape generator. The interpolated shapes are smoother and more shape-preserving, leading to a better shape space .

### 4.3 CYCLE-CONSISTENCY REGULARIZATION LOSS

The induced correspondences defined in (7) enable us to compute correspondences between two shapes by composing induced correspondences along a path of intermediate shapes. An additional regularization we can enforce is that the induced correspondences are cycle-consistent. To this end, we constrain 3-cycle consistency (Huang & Guibas, 2013) among three neighboring synthetic shapes $f^\phi(\boldsymbol{x}, \boldsymbol{z}) = 0$, $f^\phi(\boldsymbol{x}, \boldsymbol{z} + \epsilon\boldsymbol{v}) = 0$, and $f^\phi(\boldsymbol{x}, \boldsymbol{z} + \epsilon\boldsymbol{v}') = 0$, where $\boldsymbol{v}$ and $\boldsymbol{v}'$ are two different displacement vectors. Formally speaking, we model 3-cycle distortion as

$$\boldsymbol{r}^{\boldsymbol{v},\boldsymbol{v}'}(\boldsymbol{z}) := \boldsymbol{d}^{\boldsymbol{v}}(\boldsymbol{z}) + \boldsymbol{d}^{\boldsymbol{v}'-\boldsymbol{v}}(\boldsymbol{z} + \epsilon\boldsymbol{v}) - \boldsymbol{d}^{\boldsymbol{v}'}(\boldsymbol{z}) \approx -\epsilon^2\left(\boldsymbol{v}^T \frac{\partial G^\phi(\boldsymbol{z})}{\partial \boldsymbol{z}}\right)(\boldsymbol{v} - \boldsymbol{v}'). \tag{10}$$

Based on (10), we define the cycle-consistency regularization term as

$$r_{\text{cyc}}(f^\phi) = \mathbb{E}_{\boldsymbol{z} \sim \mathcal{N}_d} \int_{\boldsymbol{v} \in \mathcal{B}^d} \|\frac{\partial G^\phi(\boldsymbol{z})}{\partial \boldsymbol{z}}\|^2_{\mathcal{F}} \cdot d\boldsymbol{v} \tag{11}$$

where $\| \cdot \|_{\mathcal{F}}$ is the tensor Frobenus norm. We use finite-difference to compute $r_{\text{cyc}}(f^\phi)$. Specifically, we compute $\frac{1}{\epsilon}\|G^\phi(\boldsymbol{z} + \epsilon_{\text{cyc}}\boldsymbol{e}_i) - G^\phi(\boldsymbol{z})\|^2_{\mathcal{F}}$ as an approximation of $r_{\text{cyc}}(f^\phi)$, where $\boldsymbol{e}_i$ is a random standard basis in $\mathbb{R}^d$. In Section 7, we quantitatively show that $r_{\text{cyc}}(f^\phi)$ further enhances the shape space.

### 4.4 IMPLEMENTATION DETAILS

We use the VAE network proposed in SALD (Atzmon & Lipman, 2021), where the encoder $h^\psi$ is a modified PointNet (Qi et al., 2017) and the decoder $f^\phi$ is an 8-layer MLP. The data loss $l_{\text{VAE}}$ is the VAE loss of SALD. We set $\lambda_{\text{geo}} = 1e^{-3}$, $\lambda_{\text{cyc}} = 1e^{-4}$, and $\epsilon = 1e^{-3}$. We use autograd in PyTorch (Paszke et al., 2019) to compute $F^\phi(\boldsymbol{z})$ and $C^\phi(\boldsymbol{z})$. For other derivative computations, we use finite-difference for approximation. More details are deferred to the supp. material.

## 5 STAGE II: MESH GENERATOR INITIALIZATION

The second stage initializes the mesh generator $\boldsymbol{m}^\theta$ using the implicit shape generator $f^\phi$ obtained in the previous stage. GenCorres uses the same mesh generator as ARAPReg (Huang et al., 2021), which maps the latent code $\boldsymbol{z}$ to displacement vectors associated with vertices of a template mesh $\mathcal{M}$. We use the learned encoder $h^\psi$ to find the latent code $\boldsymbol{z}_{\text{temp}}$ of $\mathcal{M}$. Let $\boldsymbol{z}_i = h^\psi(S_i)$ be the latent code of the input shape $S_i$. We generate $T$ intermediate shapes $\boldsymbol{g}^\phi(\boldsymbol{z}_i^j), 1 \leq j \leq T$ ($T = 10$ in our experiments) by linearly interpolating $\boldsymbol{z}_{\text{temp}}$ and $\boldsymbol{z}_i$: $\boldsymbol{z}_i^j = \boldsymbol{z}_{\text{temp}} + j\frac{\boldsymbol{z}_i - \boldsymbol{z}_{\text{temp}}}{T+1}$. We then apply non-rigid registration to align the template mesh $\mathcal{M}$ with each intermediate shape $\boldsymbol{g}^\phi(\boldsymbol{z}_i^j)$ in order, i.e., the alignment of one intermediate shape provides the initialization for aligning the next intermediate shape. Non-rigid alignment adopts an ARAP deformation energy, and the details are deferred to the supp. material.

After propagating the correspondences along the interpolation path in the shape space, we obtain the deformed template $\boldsymbol{m}_i^{\text{init}}$ for each input shape $S_i$. We then initialize the mesh generator $\boldsymbol{m}^\theta$ using the standard regression loss:

$$\theta^{\text{init}} = \underset{\theta}{\arg\min} \sum_{i=1}^{n} \|\boldsymbol{m}_i^{\text{init}} - \boldsymbol{m}^\theta(\boldsymbol{z}_i)\|^2. \tag{12}$$

## 6 STAGE III: MESH GENERATOR REFINEMENT

The third stage refines the mesh generator $\boldsymbol{m}^\theta(\boldsymbol{z})$ by solving (2). To this end, we define the distance between the mesh generator and the input shape collection as

$$d_{\text{exp}}(\boldsymbol{m}^\theta, \overline{\mathcal{S}}) := \frac{1}{n} \sum_{i=1}^{n} l_{\text{CD}}(\boldsymbol{m}^\theta(\boldsymbol{z}_i), \mathcal{S}_i), \tag{13}$$

where $\boldsymbol{m}^\theta(\boldsymbol{z}_i)$ is the $i$-th generated mesh, $l_{\text{CD}}$ is the Chamfer loss. The loss can be optimized robustly thanks to the good initialization of the mesh generator from the first two stages. As (13)

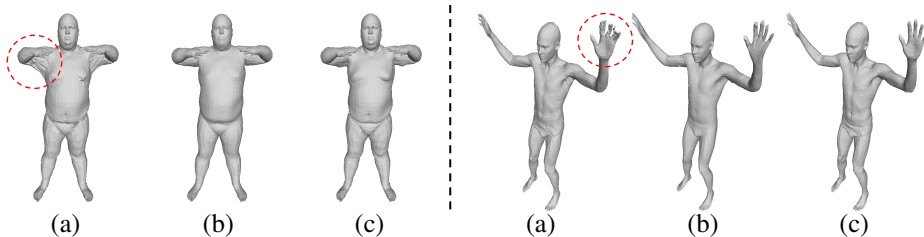

Figure 4: The mesh generator improves the inter-shape correspondences by learning better shape generation. (a) the deformed template from stage II. (b) the shape generated by the mesh generator. (c) the input raw mesh.

only constrains that vertices of the mesh generator lie on the surface, merely minimizing it does not avoid drifting. To address this issue, we define the regularization term $r_{\mathrm{d}}(\boldsymbol{m}^{\theta})$ to enforce that the deformations between meshes with similar latent codes preserve geometric structures. We enforce the deformations to be ACAP, which allows the mesh generator to capture large non-rigid deformations. Based on (5), we define

$$r_{\mathrm{d}}(\boldsymbol{m}^{\theta}) = \mathbb{E}_{\boldsymbol{z}\sim\mathcal{N}_d} \int_{\boldsymbol{v}\in\mathcal{B}^d} \boldsymbol{v}^T \frac{\partial \boldsymbol{m}^{\theta}(\boldsymbol{z})}{\partial \boldsymbol{z}}^T L^{\mathrm{acap}}(\boldsymbol{m}^{\theta}(\boldsymbol{z})) \frac{\partial \boldsymbol{m}^{\theta}(\boldsymbol{z})}{\partial \boldsymbol{z}} \boldsymbol{v} d\boldsymbol{v}. \tag{14}$$

We then substitute (13) and (14) into (2) to refine the mesh generator. As shown in Figure 4, the mesh generator can improve the shape quality from the implicit generator. Higher shape generation quality implies better inter-shape correspondences since the mesh generator directly provides consistent correspondences.

## 7 EXPERIMENTAL EVALUATION

This section presents an experimental evaluation of GenCorres. We begin with the experimental setup in Section 7.1. Section 7.2 evaluates the shape generation quality of GenCorres. We proceed to compare GenCorres with state-of-the-art JSM approaches in Section 7.3. Section 7.4 evaluate GenCorres on FAUST (Bogo et al., 2014; Ren et al., 2018). Section 7.5 presents an ablation study.

### 7.1 EXPERIMENTAL SETUP

**Datasets.** We evaluate GenCorres on two categories of deformable shape collections, i.e., Human and Animal. The Human category considers DFAUST (Bogo et al., 2017) and FAUST (Bogo et al., 2014). We use the registered SMPL model from the original DFAUST dataset. Since there is low variety between the adjacent shapes, we subsample 2000 meshes from the original dataset. For FAUST, we use the re-meshed version (Ren et al., 2018). Animal category has one dataset of 383 shapes (Huang et al., 2021), which is generated from SMAL (Zuffi et al., 2017). Due to space constraints, we defer the details of dataset processing to the supp. material.

**Evaluation protocols.** We evaluate the quality of shape generation by measuring the reconstruction errors of testing shapes, i.e., using the Chamfer distance between the reconstructed mesh and the original testing shape. For correspondence evaluation, we report the mean and median geodesic errors of the predicted correspondences between involved shape pairs.

### 7.2 EVALUATION ON SHAPE GENERATION QUALITY

We compare with the state-of-the-art shape generation approaches that do not rely on pre-defined ground-truth correspondences. Those include implicit shape generators DeepSDF (Park et al., 2019) and SALD (Atzmon & Lipman, 2021), point-based generators, such as LGF (Cai et al., 2020) and DPM (Luo & Hu, 2021). For Human category, we train the shape generator from 1000 shapes and evaluate them on the remaining 1000 shapes. For the Animal category, we use 289 shapes for training and 94 shapes for testing .

Table 1 provides quantitative comparisons between GenCorres and baseline shape generators. Figure 5 shows the qualitative results. More comparisons are in the supp. material. GenCorres is superior to all baselines in terms of both reconstruction errors and plausibility of synthetic shapes. Quantitatively, the reductions in mean/median reconstruction errors are 2.7%/10.6%, 2.3%/7.0% on DFAUST and SMAL, respectively. Qualitatively, GenCorres provides much better interpolation results compared to SALD, especially in preserving the rigidity of arms and legs of the humans. These performance gains mainly come from the geometric deformation regularization loss employed by GenCorres.

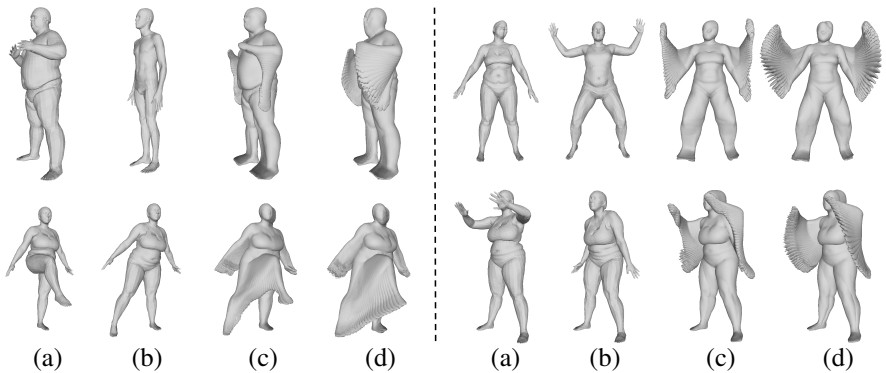

(a)        (b)        (c)        (d)          (a)        (b)        (c)        (d)

Figure 5: The comparison of shape interpolation between SALD (Atzmon & Lipman, 2021) and our method on the DFAUST dataset. (a) source shape. (b) target shape. (c) interpolation results of SALD. (d) our results.

Table 1: Evaluations of shape generation quality. For each method, we report the mean and median reconstruction errors (cm) of the testing shapes. Baselines are described in Section 7.2.

|  |  | DFAUST | | SMAL | |
|---|---|---|---|---|---|
|  |  | mean | median | mean | median |
| Point | LGF | 4.62 | 2.30 | 9.13 | 8.15 |
|  | DPM | 3.80 | 2.00 | 8.07 | 7.44 |
| Implicit | DeepSDF | 2.03 | 1.98 | 7.84 | 7.59 |
|  | SALD | 1.88 | 1.79 | 7.66 | 7.32 |
|  | GenCorres | **1.83** | **1.75** | **6.85** | **6.81** |

Table 2: Evaluations of JSM on DFAUST and SMAL using geodesic errors of the predicted correspondences (in cm). Baselines are described in Section 7.3.

|  | DFAUST | | SMAL | |
|---|---|---|---|---|
|  | mean | median | mean | median |
| CZO | 3.71 | 3.68 | 1.19 | 1.12 |
| MIM | 3.42 | 3.40 | 1.30 | 1.29 |
| NeuroMorph | 2.49 | 2.47 | 1.59 | 1.43 |
| GenCorres | **1.30** | **1.13** | **1.02** | **0.47** |
| GenCorres-NoCycle | 1.41 | 1.22 | 1.11 | 0.51 |
| GenCorres-NoGeoReg | 7.65 | 7.34 | 6.28 | 5.09 |
| GenCorres-NoACAP | 1.62 | 1.42 | 1.07 | 0.49 |
| GenCorres-Imp | 2.62 | 1.81 | 1.24 | 0.53 |

## 7.3 Evaluation on Joint Shape Matching

Table 2 reports statistics of GenCorres for JSM on DFAUST and SMAL. For baseline comparison, we choose consistent zoom out (CZO) (Huang et al., 2020a) and multiple isometric matching (MIM) (Gao et al., 2021), which are two state-of-the-art JSM approaches. We evaluate the methods by computing the correspondence error between a template shape to rest of the shapes. We also report the performance of the top-performing pair-wise matching approach NeuroMorph (Eisenberger et al., 2021) on these pairs. Note that NeuroMorph is originally not designed for JSM problem. Overall, GenCorres outperforms both JSM baselines by large margins. Specifically, GenCorres reduces the mean/median errors by 62.0%/66.8% and 14.3%/58.0% on DFAUST and SMAL, respectively. The performance gains come from two aspects. First, enforcing ARAP and ACAP deformations in the shape space locally is superior to applying sophisticated deformation models between pairs of shapes directly. Second, GenCorres performs map synchronization on synthetic shapes of the generator whose size is much larger than the input shape collection used by JSM baselines.

## 7.4 Evaluation on Pair-wise Shape Matching

Most of the shape matching approaches are evaluated on the pairwise benchmark FAUST (Bogo et al., 2014) with 20 testing shapes. We apply JSM on these shapes using GenCorres. However,

Table 3: Evaluations of pair-wise matching on FAUST dataset (in cm). Baselines are described in Section 7.4. Both NM and our method apply nonrigid registration as a post-processing step.

|  | Axiomatic | | | Spectral Learning | | | | | Template Based | | |
|---|---|---|---|---|---|---|---|---|---|---|---|
|  | BCICP | ZO | S-Shells | GeoFM | AFmap | D-Shells | ULRSSM | NM | 3D-CODED | IFS | Ours |
| error | 6.4 | 6.1 | 2.5 | 1.9 | 1.9 | 1.7 | **1.6** | **1.6** | 2.5 | 2.6 | **1.6** |

directly applying it on 20 input shapes of FAUST does not offer satisfactory results since learning a deformable shape space from few shapes is very difficult. To show the advantage of GenCorres, we augment the input shapes with unorganized shapes from FAUST and DFAUST dataset, resulting in 1100 shapes in total. The inter-shape correspondences between two input shapes are given by the correspondences induced from the template model.

Quantitative results are shown in Table 3. We mainly compare with state-of-the-art template based approaches, including 3D-CODED (Groueix et al., 2018) and IFS (Sundararaman et al., 2022). For completeness, we also provide the results of the axiomatic methods, including BCICP (Ren et al., 2018), ZO (Melzi et al., 2019), and S-Shells (Eisenberger et al., 2020a); and the spectral learning methods, including GeoFM (Donati et al., 2020), AFmap (Li et al., 2022), ULRSSM (Cao et al., 2023), D-Shells (Eisenberger et al., 2020b) and NM (Eisenberger et al., 2021). Note that template based methods do not utilize intrinsic features, thus they usually have worse performance compared to spectral learning methods, especially in the region of self-intersection. GenCorres (Ours) outperforms all template based methods. It also achieves comparable performances to spectral learning methods. How to incorporate intrinsic features into our pipeline is left for future research.

## 7.5 Ablation Study

This section presents an ablation study on different components of GenCorres. As the main purpose of GenCorres is inter-shape correspondences, we focus on how the correspondence quality changes when varying different components of GenCorres (See Table 2).

**Without the cycle-consistency regularization.** Dropping this term hurts the implicit generator. Quantitatively, the correspondence errors increase by 8.4%/7.9% and 8.8%/8.5% in mean/median on DFAUST and SMAL.

**Without the geometric regularization.** The performance of GenCorres drops considerably when removing the geometric deformation regularization term. The mean/median geodesic errors increase by 488%/549% and 515%/982% on DFAUST and SMAL. This shows that even the cycle-consistency constraint is enforced on the correspondences computed from optimizing ARAP and ACAP losses, constraining that these correspondences minimize ARAP and ACAP losses is critical.

**ACAP versus ARAP.** GenCorres-NoACAP replaces ACAP regularization with the ARAP regularization. As shown in Table 2, the performance of GenCorres slightly decreases. In particular, on DFAUST that exhibit large inter-shape deformations, i.e., thin versus fat and low versus tall, the performance drops are noticeable. Such performance gaps show that the ACAP regularization loss is important for modeling large non-isometric inter-shape deformations.

**No explicit generator.** Finally, we drop the explicit generator and use the implicit shape generator to propagate correspondences computed along linearly interpolated intermediate shapes, i.e., GenCorres-Imp. The correspondences of the explicit generator (GenCorres) are superior to propagated correspondences of the implicit generator, i.e., 50.4%/37.6% and 17.7%/11.3% of error reductions on DFAUST and SMAL, respectively. Such improvements are expected as propagated correspondences between shapes that undergo large deformations may drift.

## 8 Conclusions, Limitations, and Future Work

This paper shows that learning shape generators from a collection of shapes leads to consistent inter-shape correspondences that considerably outperform state-of-the-art JSM approaches. The key novelties of GenCorres are the idea of using a mesh generator to formulate JSM and two regularization losses that enforce geometric structures are preserved and induced correspondences are cycle-consistent. We present extensive experimental results to justify the effectiveness of these two regularization terms. Besides high-quality inter-shape correspondences, GenCorres also outperforms state-of-the-art deformable shape generators trained from unorganized shape collections.

One limitation of GenCorres is that it requires a reasonably large input shape collection to learn the shape generator and does not work with few input shapes. In this latter regime, learning pairwise matching has the advantage over GenCorres. This issue may be partially addressed by using a more advanced implicit generator for deformable shapes, which is an area for future research.

There are ample future directions. So far, the regularization terms are based on discretizing implicit surfaces into meshes. An interesting question is how to define them without mesh discretization. Another direction is to explore regularization terms for man-made shapes, e.g., to enhance topological generalization and promote physical stability.

ACKNOWLEDGMENTS

We would like to acknowledge NSF IIS-2047677, HDR-1934932, and CCF-2019844.

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

# Appendices

The supplementary materials provide more details of implementing the regularization loss used in the implicit shape generator in Section A and details of mesh generator initialization in Section B. Section C gives more details on the dataset preprocessing. Section D and Section E show more results on shape space learning and shape matching, respectively.

## A  DETAILS OF REGULARIZATION LOSS

### A.1  EXPRESSION OF $\overline{L}^{\text{arap}}(\boldsymbol{g}^{\phi}(\boldsymbol{z}))$

$$\overline{L}^{\text{arap}}(\boldsymbol{g}^{\phi}(\boldsymbol{z})) = 2L \otimes I_3 - B^{\text{arap}}(\boldsymbol{g}^{\phi}(\boldsymbol{z})) D^{\text{arap}}(\boldsymbol{g}^{\phi}(\boldsymbol{z})) B^{\text{arap}}(\boldsymbol{g}^{\phi}(\boldsymbol{z}))^T,$$

where $L$ is the graph Laplacian of the mesh, and $B^{\text{arap}}(\boldsymbol{g}^{\phi}(\boldsymbol{z})) \in \mathbb{R}^{3n \times 3n}$ is a sparse block matrix defined as

$$B_{ij}^{\text{arap}}(\boldsymbol{g}^{\phi}(\boldsymbol{z})) = \begin{cases} \sum\limits_{k \in \mathcal{N}_i} \boldsymbol{e}_{ik}^{\phi}(\boldsymbol{z}) \times & i = j \\ \boldsymbol{e}_{ij}^{\phi}(\boldsymbol{z}) \times & j \in \mathcal{N}_i \\ 0 & \text{else} \end{cases}$$

where $\boldsymbol{e}_{ij}^{\phi}(\boldsymbol{z}) = \boldsymbol{g}_i^{\phi}(\boldsymbol{z}) - \boldsymbol{g}_j^{\phi}(\boldsymbol{z})$ and $D^{\text{arap}}(\boldsymbol{g}^{\phi}(\boldsymbol{z})) \in \mathbb{R}^{3n \times 3n}$ is a diagonal block matrix defined as

$$D_{ii}^{\text{arap}}(\boldsymbol{g}^{\phi}(\boldsymbol{z})) = \Big( \sum_{j \in \mathcal{N}_i} \big( \|\boldsymbol{e}_{ij}^{\phi}(\boldsymbol{z})\|^2 I_3 - \boldsymbol{e}_{ij}^{\phi}(\boldsymbol{z}) \boldsymbol{e}_{ij}^{\phi}(\boldsymbol{z})^T \big) \Big)^{-1}$$

### A.2  EXPRESSION OF $\overline{L}^{\text{acap}}(\boldsymbol{g}^{\phi}(\boldsymbol{z}))$

$$\overline{L}^{\text{acap}}(\boldsymbol{g}^{\phi}(\boldsymbol{z})) = 2L \otimes I_3 - B^{\text{acap}}(\boldsymbol{g}^{\phi}(\boldsymbol{z})) D^{\text{acap}}(\boldsymbol{g}^{\phi}(\boldsymbol{z})) B^{\text{acap}}(\boldsymbol{g}^{\phi}(\boldsymbol{z}))^T,$$

where $B^{\text{acap}}(\boldsymbol{g}^{\phi}(\boldsymbol{z})) \in \mathbb{R}^{3n \times 4n}$ is a sparse block matrix defined as

$$B_{ij}^{\text{acap}}(\boldsymbol{g}^{\phi}(\boldsymbol{z})) = \begin{cases} \sum\limits_{k \in \mathcal{N}_i} \big( -\boldsymbol{e}_{ik}^{\phi}(\boldsymbol{z}) \quad \boldsymbol{e}_{ik}^{\phi}(\boldsymbol{z}) \times \big) & i = j \\ \big( -\boldsymbol{e}_{ij}^{\phi}(\boldsymbol{z}) \quad \boldsymbol{e}_{ij}^{\phi}(\boldsymbol{z}) \times \big) & j \in \mathcal{N}_i \\ 0 & \text{else} \end{cases}$$

and $D^{\text{acap}}(\boldsymbol{g}^{\phi}(\boldsymbol{z})) \in \mathbb{R}^{4n \times 4n}$ is a diagonal block matrix defined as

$$D_{ii}^{\text{acap}}(\boldsymbol{g}^{\phi}(\boldsymbol{z})) = \Big( \sum_{j \in \mathcal{N}_i} \big( \|\boldsymbol{e}_{ij}^{\phi}(\boldsymbol{z})\|^2 I_4 - \text{diag}(0, \boldsymbol{e}_{ij}^{\phi}(\boldsymbol{z}) \boldsymbol{e}_{ij}^{\phi}(\boldsymbol{z})^T) \big) \Big)^{-1}$$

### A.3  EXPRESSION OF $C^{\phi}(\boldsymbol{z})$ AND $F^{\phi}(\boldsymbol{z})$

Considering all vertices, the matrix representation of

$$\frac{\partial f^{\phi}}{\partial \boldsymbol{x}}(\boldsymbol{g}_i^{\phi}(\boldsymbol{z}), \boldsymbol{z})^T \boldsymbol{d}_i^{\boldsymbol{v}}(\boldsymbol{z}) + \epsilon \frac{\partial f^{\phi}}{\partial \boldsymbol{z}}(\boldsymbol{g}_i^{\phi}(\boldsymbol{z}), \boldsymbol{z})^T \boldsymbol{v} = 0.$$

can be written as $C^{\phi}(\boldsymbol{z})\boldsymbol{d} = -\epsilon F^{\phi}(\boldsymbol{z})\boldsymbol{v}$, where

$$\boldsymbol{d} = \begin{bmatrix} \boldsymbol{d}_1^{\boldsymbol{v}}(\boldsymbol{z}) \\ \boldsymbol{d}_2^{\boldsymbol{v}}(\boldsymbol{z}) \\ \vdots \\ \boldsymbol{d}_n^{\boldsymbol{v}}(\boldsymbol{z}) \end{bmatrix} \in \mathbb{R}^{3n},$$

$$C^\phi(\boldsymbol{z}) = \begin{bmatrix} \frac{\partial f^\phi}{\partial \boldsymbol{x}}(\boldsymbol{g}_1^\phi(\boldsymbol{z}), \boldsymbol{z})^T & & & \\ & \frac{\partial f^\phi}{\partial \boldsymbol{x}}(\boldsymbol{g}_2^\phi(\boldsymbol{z}), \boldsymbol{z})^T & & \\ & & \ddots & \\ & & & \frac{\partial f^\phi}{\partial \boldsymbol{x}}(\boldsymbol{g}_n^\phi(\boldsymbol{z}), \boldsymbol{z})^T \end{bmatrix} \in \mathbb{R}^{n \times 3n},$$

$$F^\phi(\boldsymbol{z}) = \begin{bmatrix} \frac{\partial f^\phi}{\partial \boldsymbol{z}}(\boldsymbol{g}_1^\phi(\boldsymbol{z}), \boldsymbol{z})^T \\ \frac{\partial f^\phi}{\partial \boldsymbol{z}}(\boldsymbol{g}_2^\phi(\boldsymbol{z}), \boldsymbol{z})^T \\ \vdots \\ \frac{\partial f^\phi}{\partial \boldsymbol{z}}(\boldsymbol{g}_n^\phi(\boldsymbol{z}), \boldsymbol{z})^T \end{bmatrix} \in \mathbb{R}^{n \times d},$$

### A.4 Implementation Details

Both the geometric deformation regularization $r_{\text{geo}}(f^\phi)$ and the cycle-consistency regularization $r_{\text{cyc}}(f^\phi)$ rely on the mesh with $n$ vertices that is discretized from $f^\phi(\boldsymbol{x}, \boldsymbol{z}) = 0$. We use Marching Cube for discretization. For the human category, we use a voxel grid with size $64 \times 77 \times 64$. For the animal category, the size of the voxel grid is $82 \times 50 \times 71$. The output mesh from the Marching Cube algorithm typically contains more than 5000 vertices. To reduce the computation complexity, we simplify the output mesh into 2000 faces (Garland & Heckbert, 1997) before computing $r_{\text{geo}}(f^\phi)$ and $r_{\text{cyc}}(f^\phi)$. The number of vertices $n$ is around 1000, thus the size of $\overline{L}^\phi(\boldsymbol{z})$ is around $3000 \times 3000$. Computing $(\overline{L}^\phi(\boldsymbol{z}))^+$ only takes about 40ms in PyTorch (Paszke et al., 2019)

## B Details of Mesh Generator Initialization

### B.1 Template-based Registration

In order to register the template mesh $\mathcal{M}$ to the input shape $S_i$, we first generate $T$ intermediate shape $\boldsymbol{g}^\phi(\boldsymbol{z}_i^j)$, where $\boldsymbol{z}_i^j = \boldsymbol{z}_{\text{temp}} + j\frac{\boldsymbol{z}_i - \boldsymbol{z}_{\text{temp}}}{T+1}, 1 \leq j \leq T$. Instead of directly register $\mathcal{M}$ to $S_i$, we first register $\mathcal{M}$ to $\boldsymbol{g}^\phi(\boldsymbol{z}_i^1)$ with ARAP deformation energy (Sorkine & Alexa, 2007; Huang et al., 2008). Since $\mathcal{M}$ and $\boldsymbol{g}^\phi(\boldsymbol{z}_i^1)$ are very close, we directly apply nearest neighbor search to compute the correspondence for the data term. The registration gives the resulting deformed template $\mathcal{M}_1$. We then register $\mathcal{M}_1$ to $\boldsymbol{g}^\phi(\boldsymbol{z}_i^2)$ and get the deformed template $\mathcal{M}_2$, register $\mathcal{M}_2$ to $\boldsymbol{g}^\phi(\boldsymbol{z}_i^3)$ and get the deformed template $\mathcal{M}_3$, and so on and so forth. Finally we get the deformed template $\mathcal{M}_m$, which is well-aligned with $S_i$.

The interpolation-guided registration typically works well but might fail when the template $\mathcal{M}$ is too far from $S_i$, i.e. the two shapes have very different poses. The reason is that the intermediate shapes on the interpolation path might not have good quality. The make full use of the learned shape space, we add shapes from the input shape collection to the interpolation path in these cases. First, we compute the distance between each pair of shape $S_i$ and $S_j$ using the distance of their embedded latent codes $\|\boldsymbol{z}_i - \boldsymbol{z}_j\|$. Based on this distance metric, we build a $K$-NN graph among the input shape collection. We set $K = 25$ for the human dataset and $K = 40$ for the animal dataset. We then perform interpolation-guided registration on each edge $(i, j)$ of the graph and obtain the correspondences between $S_i$ and $S_j$. We use the distortion of the mapped edges (Huang et al., 2008) as the weights in the $K$-NN graph. Finally we compute the shortest path from the template to each shape $S_i$ and get the correspondences by composing the correspondences along the shortest path.

### B.2 Mesh Generator Architecture

The network architecture of $\boldsymbol{m}^\theta$ follows from that in (Huang et al., 2021), which outputs displacements of vertex positions of the template mesh $\mathcal{M}$. We sample 4 resolutions of the mesh connections of the template mesh. The network architecture stacks 4 blocks of convolution + up-sampling layers. The convolution layer employs Chebyshev convolutional filters with 6 Chebyshev polynomials (Ranjan et al., 2018). Similar to (Zhou et al., 2020), there is a fully connected layer between the latent code and the input to the first convolution layer.

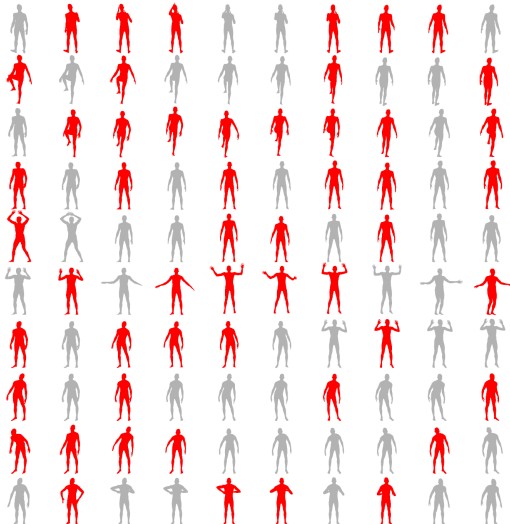

Figure 6: We use FPS to further select a more diverse and challenging subset of 1000 shapes from the training split (Atzmon & Lipman, 2021; 2020; Gropp et al., 2020). In this example, among $10 \times 10 = 100$ shapes from the training split (Atzmon & Lipman, 2021; 2020; Gropp et al., 2020), only the red shapes are selected because the gray shapes have similar poses.

## C  DETAILS OF DATASETS

There are approximately 41k shapes in the original DFAUST dataset. Since there is low variety between the adjacent shapes, recent works (Atzmon & Lipman, 2021; 2020; Gropp et al., 2020) create a new training/testing split by uniformly sample $20\%$ shapes from the original dataset. However, we notice that there are still many similar shapes in the splits. For example, almost all motion sequences start from shapes with very similar rest pose. In order to make the dataset more challenging, we further select 1000 shapes from the training split. Specifically, we first learn a VAE (Atzmon & Lipman, 2021) to embed all the shapes from the training split into different latent vectors. We find that the latent vectors of similar shapes are typically closer. Then we apply farthest point sampling (FPS) to the latent vectors and select the first 1000 shapes. An example is shown in Figure 6. We apply the same approach to the testing split and select another 1000 shapes for testing.

The original SMAL dataset from (Huang et al., 2021) contains 300 training shapes and 100 testing shapes. We filter out the shapes that have unreasonable self-intersection, leading to 289 training shapes and 94 testing shapes.

For both DFAUST and SMAL dataset, we evaluate the correspondences from a template shape to the remaining shapes.

## D  MORE RESULTS OF SHAPE SPACE LEARNING

We show the shape interpolation results of the state-of-the-art implicit generator (Atzmon & Lipman, 2021) and our method in Figure 7 and Figure 8. We show 30 interpolated shapes. By adding the proposed geometric deformation regularization loss and cycle-consistency regularization loss, our generator gives more meaningful interpolation results, which are important to the interpolation-guided registration.

## E  MORE RESULTS OF SHAPE MATCHING

We show the correspondence results of NeuroMorph (Eisenberger et al., 2021) and our method in Figure 9 and Figure 10. Our method has lower errors compared to NeuroMorph.

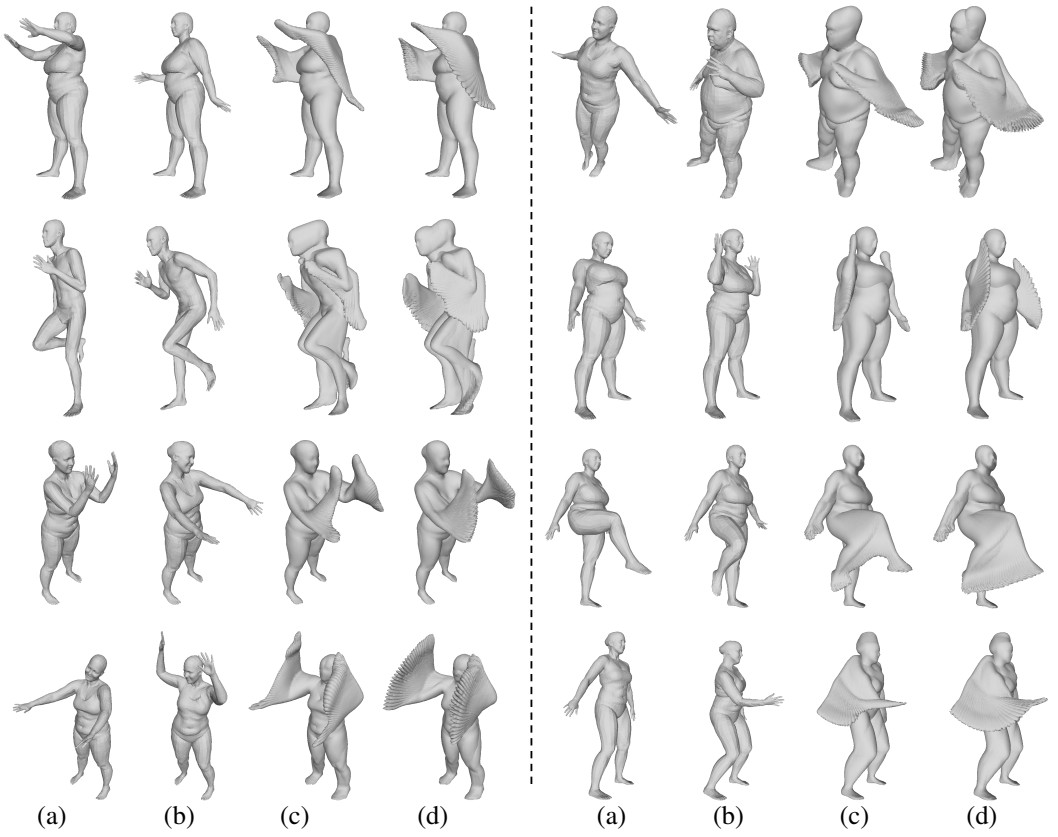

Figure 7: The comparison of shape interpolation between SALD (Atzmon & Lipman, 2021) and our method on the DFAUST dataset. (a) source shape. (b) target shape. (c) interpolation results of SALD (Atzmon & Lipman, 2021). (d) our results.

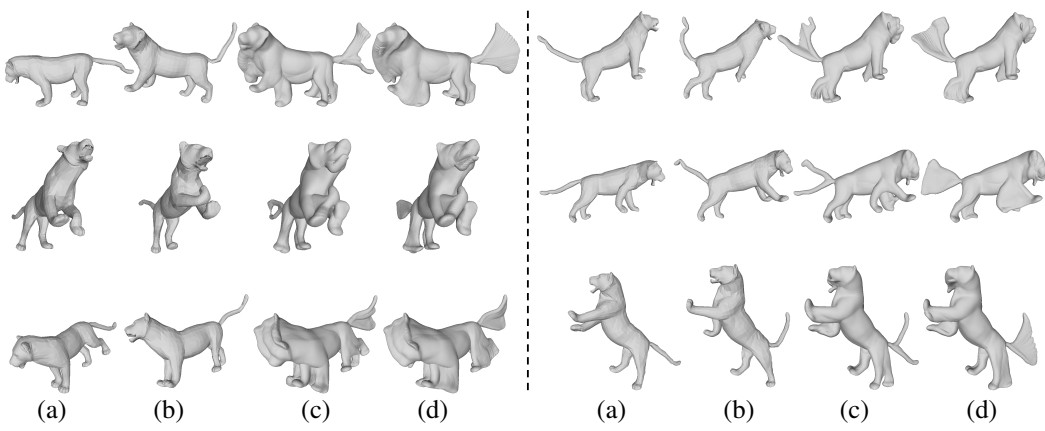

Figure 8: The comparison of shape interpolation between SALD (Atzmon & Lipman, 2021) and our method on the SMAL dataset. (a) source shape. (b) target shape. (c) interpolation results of SALD (Atzmon & Lipman, 2021). (d) our results.

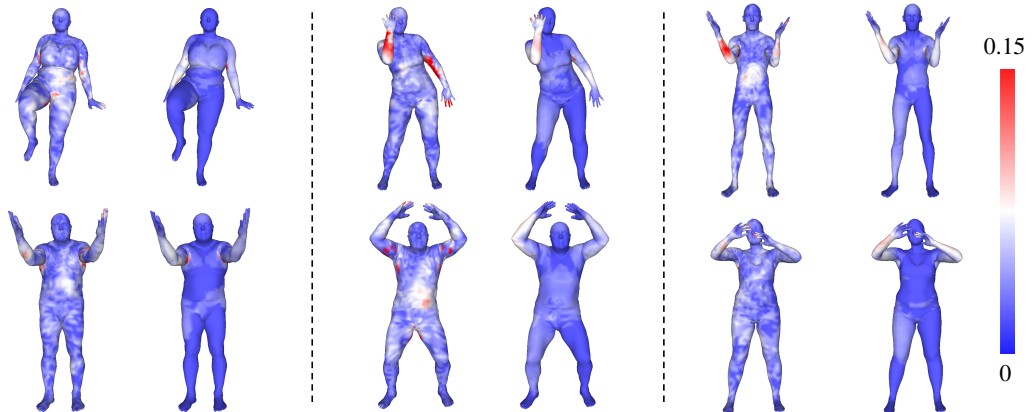

Figure 9: The comparison of the correspondences between our method and NeuroMorph (Eisenberger et al., 2021) on the DFAUST dataset. We show the correspondence errors on the target shapes. For each group of shapes, the left is from NeuroMorph, the right is our result.

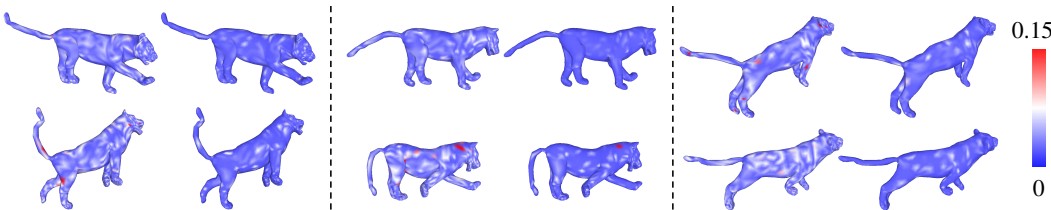

Figure 10: The comparison of the correspondences between our method and NeuroMorph (Eisenberger et al., 2021) on the SMAL dataset. We show the correspondence errors on the target shapes. For each group of shapes, the left is from NeuroMorph, the right is our result.

