# OpenReview forum: "GenCorres: Consistent Shape Matching via Coupled Implicit-Explicit Shape Generative Models"
_ICLR.cc/2024/Conference — ICLR 2024 poster_

### Official Review · Reviewer_sJwu · 2023-10-17

**Soundness:** 3 good
**Presentation:** 2 fair
**Contribution:** 3 good
**Rating:** 6
**Confidence:** 4

**Summary:**

The paper proposes joint shape matching (JSM), an unsupervised training schema based on three steps: the training of a generative model for implicit shapes; a template fitting to the shape generated by interpolation in the latent space; a final refinement provided by a chamfer fitting. The method is tested on humans and animals on relatively small datasets, and shows promising performance w.r.t. previous methods, comparable to ones that use implicit information.

**Strengths:**

- The method is unsupervised and extrinsic but provides good performance, competitive with intrinsic approaches that uses the surface as regularization
- The method seems novel, and the idea of enhancing the learning representation by interpolation in the latent space seems straightforward and reasonable

**Weaknesses:**

- The central part of the method is not easy to grasp. I find the overall principle clear, and it is probably possible to replicate the work in principle. Still, the technique is not well explained in detail, and the notation is often confusing. For example, the letter "g" is used both for the shape generator and for the mesh generator, and requires some back-and-forth to get used to it. I suggest revisiting the method explanation and clarifying the methodological details.
- By my understanding, the method highly relies on the target shapes belonging to a given distribution that should not only bounded in terms of structure and class but in particular, should be possible to express by the registered template. Many methods rely on template registration, and it is not a weakness per se, but the proposed approach makes use of techniques that aim to be general and flexible (e.g., implicit representation, unsupervised learning), but the paper does not show any out-of-distribution results (and also, results only on shapes for which data are available and can be even generated synthetically). I also believe that the topological constraint given by the template limits the generality of the method. The limitations barely mention this, and it should be emphasized more.
- Experiments are performed on a relatively small set of data. The datasets are outdated and do not deal with the literature's more recent and real challenges (e.g., partiality, noise, clutter, ...). In this sense, I suggest stressing the method further; I believe that they would be interesting to investigate the latent interpolation when the learned space represents more diverse shapes (i.e., I wonder if relying on implicit representation may help in the context of limited topological variation during the interpolation, or if the relying on a template do not let to generalize to these cases)

**Questions:**

Following the Weaknesses above:
1) How would linear interpolation behave in the presence of significant diverse geometrical topology data? Is the template triangulation the main limitation in this case?
2) Could you provide some measure of the computational cost of the method? I think would be important to understand the scalability of the method, and I assume that the method would require a significant computational effort. How does it compare with NeuroMorph?
3) Would it be possible to test the method on a class for which a dense correspondence is not provided; for example, some classes of ShapeNet (e.g., cars, airplanes)?

---

> ### Author Response · Authors · 2023-11-21
> **Response to reviewer sJwu**
>
> Thank you for your valuable comments and suggestions. Below are our clarifications for your concerns. We value your input and would love to hear any follow-up you have to the response.
>
> > **Q: Confusing notation.**
>
> In the original submission, the implicit generator uses lowercase letters $g^{\phi}$. The explicit generator uses bold letters $\mathbf{g}^{\theta}$. In the revised manuscript, we have made changes to the notations to make them clearer:
>
> - $f^{\phi}$ denotes the implicit generator, where $\phi$ are the network parameters.
>
> - $f^{\phi}(\mathbf{x}, \mathbf{z})=0$ denotes the implicit surface (i.e. the iso-surface where the signed distance value is 0).
>
> - $\mathbf{g}^{\phi}(\mathbf{z})$ denotes the discretized mesh of the implicit surface $f^{\phi}(\mathbf{x}, \mathbf{z})=0$.
>
> - $h^{\psi}$ denotes the encoder in the VAE.
>
> - $\mathbf{m}^{\theta}$ denotes the mesh generator in the third stage.
>
> We have also provided the explanations in the revised main paper.
>
> > **Q: Data is generated synthetically and does not deal with the real challenges (e.g., partiality, noise, clutter, …)**
>
> We have included additional experiments in the supplementary material, submitted during the rebuttal period, to demonstrate that our method is also applicable to real scans. We trained the implicit network on a real human dataset featuring various clothes and hairstyles. The experiments indicate that our method can still provide smooth interpolation even when the learned space represents a more diverse range of shapes. Please refer to the newly submitted supplementary material for more details. Additionally, we want to emphasize that the Human dataset we evaluated, which comes from SMPL, is sufficiently challenge for inter-shape matching.
>
> > **Q: The topological constraint given by the template limits the generality of the method.**
>
> The template is introduced for the explicit mesh generator. In fact, any mesh generator in the literature requires a template.
> The template is not used in the implicit generator, which is a key contribution of this paper. The implicit generator can already provide pair-wise correspondences between pairs of implicit surfaces, which are not constrained by topology. Specifically, given any two shapes, we can perform latent space interpolation between them and propagate the correspondences along the interpolation path using equation (7) in the main paper. This framework is applicable to any shape collection, regardless of its topology. These correspondences can be evaluated through the interpolation results shown in the paper; smoother and more rigid interpolations indicate better correspondences. In this paper, we utilize the template solely for the task of joint shape matching of deformable shapes, as they typically share similar topology and structure.
>
> > **Q: How would linear interpolation behave in the presence of significant diverse geometrical topology data? Is template triangulation the main limitation in this case?**
>
> In this paper, we focus on deformable shapes (humans and animals), which usually do not have significant topological changes. For other types of shape collections with significantly diverse geometrical topologies, we leave this for future research. Note that we perform linear interpolation for the implicit surfaces, where template triangulation is not involved.
>
> > **Q: Would it be possible to test the method on a class for which a dense correspondence is not provided; for example, some classes of ShapeNet (e.g., cars, airplanes)?**
>
> We propose a general framework for computing correspondences between implicit surfaces. We are encouraged by your recognition of our framework's potential applicability to a wide range of shape collections, including man-made shapes. For man-made shapes, we can adopt alternative energies, such as as-affine-as-possible (AAAP) energy, which enforces local region transformations to be affine rather than rigid, in place of ACAP energy. However, we want to underscore that deformations among man-made shapes are more complex than deformable shape collections. Also there are structural variations, where AAAP should be enforced among shapes or parts that are structurally similar. Those issues should be addressed by other papers (we are currently looking into this).
>
> > **Q: Computational cost of the method compared with NeuroMorph.**
>
> Similar to SALD, we train our model with 8 NVIDIA TITAN V GPUs (12G). For the Human collections (about 1000 shapes), training the implicit generator takes about 3 days, while the explicit generator requires approximately 12 hours. For the FAUST dataset (80 shapes), NeuroMorph is trained with 1 GPU for roughly 3 days. Note that NeuroMorph requires a pair of shapes as input, so the training complexity is $O(n^2)$ for $n$ training shapes. In contrast, our method directly learns the shape space, meaning the complexity scales linearly with the number of training shapes.

---

> > ### Comment · Reviewer_sJwu · 2023-11-22
> > **Post Rebuttal Reply**
> >
> > I thank the Authors for their effort in answering my concerns. The additional experiments address some of my concerns, and the answers are mainly satisfactory. I do not have further questions, and I am willing to raise my score.
> > I am looking forward to hearing from other reviewers their opinion.

---

> > > ### Author Response · Authors · 2023-11-23
> > > **Response**
> > >
> > > Thank you again for your feedback!

---

### Official Review · Reviewer_pFHc · 2023-10-19

**Soundness:** 3 good
**Presentation:** 2 fair
**Contribution:** 3 good
**Rating:** 5
**Confidence:** 3

**Summary:**

The paper presents a method for computing correspondences between target shapes from a collection of shapes. Namely, the method first learns an implicit shape generator from the input collection, enforcing several useful shape-preserving constraints. It then evaluates the learned implicit generator to interpolate between target shapes, and uses the interpolation to guide the learning of the explicit shape generator. After a final refinement step, the target shapes are corresponded with high quality and appropriate in-between interpolations. The main shortcoming of the approach is the need for a substantially large collection of shapes to begin with.

**Strengths:**

The high level idea is simple and sound: learn a shape generator with useful geometric priors, then apply its interpolation capabilities to guide explicit correspondence. The results are convincing, with rigid limbs bending appropriately between corresponded shapes, and the ablation study is informative.

**Weaknesses:**

The paper is hard to follow, both in text and in math, which makes it difficult to clearly understand and implement the presented ideas. A few rounds of editing should help the flow. Much of the mathematical detail (especially in section 4.1) should be relegated to the appendices (with full derivations and descriptions), and only clearly readable final formulas should be shown in the main text (I'd carefully pick which symbols represent which concepts).

The demonstrated use case is pretty contrived -- all collections come from well parameterized datasets. I would love to see results using 3D body scans with various clothes, hairstyles, etc.

The human shape collection seems to include a multitude of body types, and is clearly able to correspond/interpolate between them. The animal collections seem to only have shapes of the same animal. The paper would be stronger if you show an interpolation between say a lion and a horse. It is not clear whether this method would blend well between very different quadruped shapes.

A more suitable ARAP reference should be: As-rigid-as-possible shape interpolation by Alexa et al. 2000

**Questions:**

The main title in the PDF seems to be misspelled: "CenCorres" instead of "GenCorres".

In general, there are many symbols used in the mathematical notation, which need to be clearly mentioned and described.
  * What are the different fonts for g (looks like implicit expressions use lowercase letters, while explicit use bold letters)
  * What is Theta in g^Theta in the "Problem statement" paragraph (page 3)?
  * What is Phi in "Approach overview" (page 3) and what does it map from and to? Why R^3 x Z? Does it compute the distance field value at the specific 3D coordinate for the shape defined by a latent code?
  * What is Psi in h^Psi in equation (1)? Also mention lambdas for equations (1) and (2).
  * Symbol x is introduced in section 4.1 without describing what it is.
  * What does d stand for in equation 3, some displacement between a vertex and an infinitesimally close corresponding vertex? Is d of unit length, then scaled down by epsilon? Or should d be the small displacement that doesn't need to be further multiplied by epsilon?
  * At some point we get exposed to several capital letter symbols (C, F, G, E) whose meaning is hard to follow.

Table 2 caption should mention that the metric is mean and median geodesic distance between correspondences.

Acronyms should be spelled out the first usage.
  * First mention of ACAP is not accompanied by description and reference, maybe just remove it from section 3 until it's properly introduced in section 4.
  * MP-pseudo inverse should just be Moore-Penrose inverse.

---

> ### Author Response · Authors · 2023-11-21
> **Response to reviewer pFHc**
>
> Thank you for your valuable comments and suggestions. Below are our clarifications for your concerns. We value your input and are keen to discuss any further questions or comments you may have following our response.
>
> > **Q: Math notations and what are the different fonts for g?**
>
> In the original submission, the implicit generator uses lowercase letters because the signed distance is a scalar. The explicit generator uses bold letters because the output vertices are vectors. They can also be distinguished by the network parameters: $\phi$ for the implicit generator and $\theta$ for the explicit generator. We used g because both of them are generative models. In the revised manuscript, we have made changes to the notations to make them clearer:
>
> - $f^{\phi}$ denotes the implicit generator, where $\phi$ are the network parameters. Yes, the reason for $\mathbb{R}^3 \times \mathcal{Z}$ is that it computes the signed distance value at the specific 3D coordinate for the shape defined by a latent code.
>
> - $f^{\phi}(\mathbf{x}, \mathbf{z})=0$ denotes the implicit surface (i.e. the iso-surface where the signed distance value is 0). $\mathbf{x}$ is the 3D query location and $\mathbf{z}$ is the latent code.
>
> - $\mathbf{g}^{\phi}(\mathbf{z})$ denotes the discretized mesh of the implicit surface $f^{\phi}(\mathbf{x}, \mathbf{z})=0$.
>
> - $h^{\psi}$ denotes the encoder in the VAE, where $\psi$ are the network parameters.
>
> - $\mathbf{m}^{\theta}$ denotes the mesh generator in the third stage, where $\theta$ are the network parameters.
>
> We have also provided these explanations in the revised main paper.
>
> > **Q: What does d stand for in equation 3, some displacement between a vertex and an infinitesimally close corresponding vertex?**
>
> Yes, $\mathbf{d}_i^{\mathbf{v}}(\mathbf{z}) $  in equation (3) denotes the displacement between a vertex and an infinitesimally close corresponding vertex. In the original submission, we further multiplied it by $\epsilon$ because this simplified equation (6), (7) and (8) by avoiding the constant $\epsilon$. However, you are correct that this is not necessary. In the revised submission, we have followed your suggestions and did not scale it by $\epsilon$. The final expressions of the regularization terms remain the same.
>
> > **Q: Explanation of capital letter symbols (C, F, G, E).**
>
> We provide detailed expressions of $C$ and $F$ in the appendices of the main paper. $C$ and $F$ are derived from the matrix representation of Equation (3).
>
> The expression of $G$ is given in equation (7). It is just a component of $\mathbf{d}_i^{\mathbf{v}}(\mathbf{z})$ that is independent of the perturbation direction $\mathbf{v}$.
>
> $E$ is derived in equation (8) and serves as a metric to measure the local rigidity and local conformality distortions between an implicit surface and its perturbation. The expression of $E$ can also be found in equation (8) and, like $G$, is independent of the perturbation direction $\mathbf{v}$.
>
> > **Q: Results using 3D body scans with various clothes, hairstyles, etc.**
>
> We have included additional qualitative results in the supplementary material submitted during the rebuttal period. We trained the implicit network on a new human dataset featuring various clothes and hairstyles. Our proposed regularizations again consistently improve the baseline network, SALD. Please refer to the newly submitted supplementary material for more details.
>
> > **Q: Interpolation between a lion and a horse.**
>
> We have included more qualitative results in the supplementary material submitted during the rebuttal period. We trained the implicit network on a new quadruped shape dataset. Visualization of the interpolation results show that our method transitions smoothly between a lion and a horse. Please refer to the newly submitted supplementary material for more details.
>
> > **Q: Citation, acronyms and typos.**
>
> We have added the new ARAP reference and corrected the acronyms and typos in the title.

---

> > ### Comment · Reviewer_pFHc · 2023-11-22
> > **Raising the rating**
> >
> > Thank you for the rebuttal and the additional results. As as result I am happy to raise my rating. A few more comments:
> >
> >
> > You are missing a closed parentheses in Equations 4 and 5 just before the L2-norm closing.
> >
> > It would be even more impactful if you blend between one pose of the lion and a very different pose of the horse (e.g. rearing).
> >
> > Since correspondence is a key output of your method, you could enhance the visualization by texture-mapping the initial mesh (e.g. with a color-coded grid) and showing where that texture map moves to in the corresponded final mesh.

---

> > > ### Author Response · Authors · 2023-11-23
> > > **Response**
> > >
> > > Thank you again for your feedback.
> > >
> > > - We have corrected the typos in Equations 4 and 5 in the updated main paper.
> > >
> > > - In the appendices, Figure 8 demonstrates interpolation between animals with very different poses. Importantly, the first row of Figure 5 in the original main paper already showcases interpolations that incorporate not only pose variations but also significant shape differences, exemplified by transitions from a very fat man to a very thin man.
> > >
> > > - Regarding the correspondence visualization, we found that an error map is a more effective way to illustrate differences for comparison purposes. This is because the error is relatively low and the texture mapping usually results in visually similar outputs.

---

### Official Review · Reviewer_MqJa · 2023-10-31

**Soundness:** 3 good
**Presentation:** 3 good
**Contribution:** 3 good
**Rating:** 8
**Confidence:** 3

**Summary:**

This paper presents GenCorres, a method to solve the Joint Shape Matching (JSM) problem for a collection of unorganized shapes.  It is based on fitting the input shapes with a mesh generator that is constrained to preserve the local structure (conformal or isometric) of the shapes. Particularly, GenCorres first learns an implicit generator from the input shapes using a VA, which produces intermediate shapes between arbitrary pairs. The paper introduces a novel approach for computing correspondences between adjacent implicit surfaces, which is used to regularize the implicit generator. Second, synthetic shapes generated by the implicit generator guide the initial fittings (template-based deformations) for learning the mesh generator. Experimental results demonstrate that GenCorres outperforms state-of-the-art JSM techniques in collections of articulated body shapes.

**Strengths:**

GenCorres is a well-crafted method divided into three stages. Stage 1 learns an implicit generator from unorganized sets using a VAE, where local (ACAP) and cycle consistency is imposed between surfaces that are close along the embedding dimension, defining the geometric deformation and cycle-consistency regularization losses. Stage 2 learns the explicit mesh generator from the VAE encoder-decoder. Stage 3 refines and fits the mesh generator to the input set of meshes and enforces again local structure consistency with ACAP. The three stages are well constructed and contain novel contributions to this field, especially the regularization losses in Stage 1. The results in the paper reveal that the GenCorres' generative method produces high-quality meshes and the correspondences across the input set significantly improve the state-of-the-art.
In terms of the quality, the methodology is well described and motivated. The results offer enough evidence that GenCorres improves the state-of-the-art of JSM in collections of shapes describing articulated body shapes. The article also includes an ablation study to assess the importance of each individual step.
In terms of significance, this method offers an interesting solution to a difficult and open problem that has important applications in computer graphics.

**Weaknesses:**

The main weakness of GenCorres, which is revealed by the experiments and commented by the authors, is the need of a relatively large set of input shapes to learn the shape generators properly. Solving this issue is a difficult task that requires further research.

GenCorres seems to be especially suitable for a particular type of shapes (articulated body or animal shapes) to which ACAP and ARAP represent good deformation constraints. Other shape collections, such as man made objects, probably represent a challenge for this method.

**Questions:**

How well does GenCorres perform with man-made shapes or other shapes that do not correspond to articulated objects?

What is the influence of the hyperparameters (lambda and epsilon) in the final result? Are the values specified in the paper valid for other sets of input shapes?

If the number of input shapes is a critical factor, I suggest the authors to include an experiment to establish the limit number for which the method significantly degrades.

---

> ### Author Response · Authors · 2023-11-21
> **Response to reviewer MqJa**
>
> Thank you for your insightful and valuable feedback. Below are our clarifications for your concerns.
>
> > **Q: The need of a relatively large set of input shapes.**
>
> Our approach follows the stream of approaches that leverage big data (e.g. nearest neighbors and graph-based learning) to solve traditional problems. The goal is to open a new direction that has many problems to work on. For example, the approach we developed to train the implicit shape generators that preserve inter-shape deformations can be extended in another setting. As another example, training an explicit shape generator to solve the inter-shape correspondence problem can stimulate future research in this area.
>
> > **Q: How well does GenCorres perform with man-made shapes or other shapes that do not correspond to articulated objects?**
>
> We propose a general framework for computing correspondences between implicit surfaces. Specifically, we analyze the changes of the generated shape under the infinitesimal perturbation of the latent code and formulate the correspondence computation as a linearly constrained quadratic program problem (refer to Equation (6) in the main paper). We are encouraged by your recognition of our framework's potential applicability to a wide range of shape collections, including man-made shapes. For man-made shapes, the key question pertains to identifying a suitable deformation model, as inter shape deformations among man-made shapes are notably complex. One possibility is that instead of using ACAP energy, we can adopt alternative energies, such as as-affine-as-possible (AAAP) energy, which enforces local region transformation to be affine rather than rigid. Also, using L1 norm rather than the L2 norm in this setting maybe more appropriate. Another challenge is to address structural variations and only enforce AAAP among structurally similar shapes. All these questions deserve other papers to study (we are looking into this as well).
>
> > **Q: What is the influence of the hyperparameters (lambda and epsilon) in the final result? Are the values specified in the paper valid for other sets of input shapes?**
>
> We did not perform an extensive grid search for the optimal hyperparameters. However, we found our methods to be quite robust against variations in $\lambda_{geo}$, $\lambda_{cyc}$ and $\epsilon$. From our experiments, when $\lambda_{geo}$ and $\lambda_{cyc}$ are set very large, the data term increases, leading to an over-smoothing of the learned shape due to the regularization term enforcing greater similarity between shapes. We set $\epsilon$ to a small value in finite differences to achieve better gradient approximation. These hyperparameters are valid for both Human and Animal categories.
>
> > **Q: The limit number for which the method significantly degrades.**
>
> For the pair-wise shape matching experiments on FAUST, we conducted ablation studies where we gradually decreased the number of additional DFAUST shapes. The results of these studies are presented in the table below.
>
> Evaluations of pair-wise matching on FAUST (in cm).
>
> | number of DFAUST shapes | 1000 | 500 | 250 |
> |:-----------------------:|:----:|:---:|:---:|
> | error                   | 1.6  | 2.5 | 4.1 |

---

> > ### Comment · Reviewer_MqJa · 2023-11-23
> > **Author response**
> >
> > The authors have made considerable efforts to improve the paper and have satisfactorily answered my questions. Looking at the rest of the reviewer's concerns, I support accepting this paper and am willing to keep or raise my score.

---

### Official Review · Reviewer_WbPs · 2023-11-02

**Soundness:** 4 excellent
**Presentation:** 4 excellent
**Contribution:** 4 excellent
**Rating:** 8
**Confidence:** 4

**Summary:**

This paper proposes a new algorithm for joint matching of a set of 3D shapes represented as polygon meshes or surface point clouds. The core idea is to learn a generative model for meshes, based on deformations of a base template, that can produce the input shapes and thereby infer correspondences between them. To achieve this in a robust and accurate way, the authors propose several innovations, including learning an initial implicit generator that informs the eventual mesh generator, and techniques to regularize the generator output by imposing distortion-minimizing/consistency-maximizing losses in epsilon-neighborhoods of the synthesized shape space (not just around the input landmarks).

**Strengths:**

I really like this paper. I think it addresses an important, general problem by bringing together a bunch of relevant ideas in a well-justified way: using generative models for discriminative problems, physically-inspired regularization, etc. The core insights are simple and compelling. The individual technical contributions are theoretically meaningful and elegant, well-presented, and show significant improvements over baselines in experiments. I particularly like the computation of optimal correspondence fields between similar implicit shapes in 4.1. And not being an expert in this precise area, I am impressed that this can be done both differentiably (is this where finite differences come in as mentioned in 4.4?) and fast (how long does the training take?)

I am provisionally recommending acceptance and am open to revising my opinion further upwards (if the authors can address the critiques in Weaknesses) or downwards (if other reviewers find serious flaws).

**Weaknesses:**

This is a fairly complex system. I am not 100% certain that every design decision is fully justified since there are too many possible ablations (though the authors do study several obvious ones). In particular, the authors claim that the implicit approach is necessary since learning a mesh generator from scratch is too difficult and error-prone. While this may be true, it is not actually demonstrated within the ambit of the proposed pipeline. There is the 3D-CODED comparison, but that is an entirely different pipeline. I do understand that this would be an "ablation" that's at least half a research project by itself, but still... Maybe the authors have already done experiments to verify this which are not included in the paper?

Also, code (and preprocessed data) would be really helpful so that others can understand and verify the claimed contributions. Will it be released?

A couple of relevant papers may be worth mentioning since they are in the overall spirit of this paper:

Muralikrishnan et al., "GLASS: Geometric Latent Augmentation for Shape Spaces", CVPR 2022
(learning generative models from sparse landmarks guided by ARAP regularization -- does require input with correspondences though so it's sort of complementary to this paper)

Bednarik et al., "Temporally-Coherent Surface Reconstruction via Metric-Consistent Atlases", ICCV 2021
(using metric distortion energies for consistent reconstruction, and hence joint matching, of time-varying shape sequences)

Minor:
p3: Chamber --> chamfer (spelling, and should be lowercase)

**Questions:**

Please see questions inline in Strengths and Weaknesses.

---

> ### Author Response · Authors · 2023-11-21
> **Response to reviewer WbPs**
>
> Thank you for your insightful and valuable feedback. Below are our clarifications for your concerns.
>
> > **Q: This can be done both differentiably. (is this where finite differences come in as mentioned in 4.4?)**
>
> Yes, both autograd and finite differences are differentiable.
>
> > **Q: How long does the training take?**
>
> Similar to SALD, we train our model with 8 NVIDIA TITAN V GPUs (12G). For the Human collections, the training of the implicit generator takes about 3 days, while the explicit generator requires approximately 12 hours.
>
> > **Q: Ablation studies to validate the necessity of the implicit approach.**
>
> We conducted ablation studies for the joint shape matching experiments on DFAUST. The results are shown in the table below. GenCorres-NoImp does not utilize the implicit approach (1st stage) and directly registers the template to each target shape as initialization. GenCorres-NoInit omits both the 1st and 2nd stages, using only the 3rd stage. The results show that GenCorres is superior to both GenCorres-NoImp and GenCorres-NoInit, thanks to better initialization. Without latent space interpolation to generate intermediate shapes, GenCorres-NoImp often fails to register the template directly to a vastly different target shape, leading to poor initialization. GenCorres-NoInit, which does not employ any correspondence initialization, tends to get stuck in undesired local minima. Note that the success of training 3D-CODED relies on a very large dataset (23000 human meshes with a large variety of realistic poses and body shapes).
>
> Evaluations of JSM on DFAUST using geodesic errors of the predicted correspondences (in cm).
>
> |                  | mean  | median |
> |:---------------- |:-----:|:------:|
> | GenCorres        | 1.30  | 1.13   |
> | GenCorres-NoImp  | 7.92  | 7.73   |
> | GenCorres-NoInit | 12.11 | 11.84  |
>
> > **Q: Will codes and preprocessed data be released?**
>
> In the newly submitted supplementary material, we provide an additional experiment that applies the implicit generator to a new human dataset featuring various clothes and hairstyles. We have also included the corresponding code for this experiment along with a readme file. More code and data from the main paper will be released later.
>
> > **Q: Relevant papers and typos.**
>
> We have added the relevant papers and corrected the typos.

---

> > ### Comment · Reviewer_WbPs · 2023-11-23
> > **Support acceptance**
> >
> > Thanks to the authors for their responses to our questions. I supported acceptance earlier and continue to do so now (maybe with an even higher score), given that my not-super-critical concerns have been addressed, and given that other reviewers with more critical views appear to have had their concerns addressed as well.

---

### Meta-Review · Area_Chair_gSNB · 2023-12-07

**Metareview:**

This work proposes a technique for 3D shape correspondences. Majority of the reviewers lean towards accepting the work. Reviewers appreciated the well-crafted system. Several reviewers felt that the paper is difficult to follow and writing can be improved. As authors promised to release the code, it is felt that the contributions outweigh the issues and majority of the reviewers recommend the acceptance. The reviewers did raise some valuable concerns that should be addressed in the final camera-ready version of the paper, which include adding the relevant rebuttal discussions and revisions in the main paper. The authors are encouraged to make the necessary changes to the best of their ability. It is important to release the code to advance further reserach in this problem.

**Justification For Why Not Higher Score:**

Multiple reviewers mentioned that paper is somewhat hard to follow.

**Justification For Why Not Lower Score:**

Contributions outweigh the weaknesses for acceptance.

---

### Decision · Program_Chairs · 2024-01-16

Accept (poster)